# Small-scale protocols to characterize mitochondrial Complex V activity and assembly in peripheral blood mononuclear cells

**Kai-Yin Chau[1], Jan-Willem Taanman** **[1]\*, Anthony H.V. Schapira[1,2]**

**1** Department of Clinical and Movement Neurosciences, Queen Square Institute of Neurology, University College London, London, United Kingdom, **2** Aligning Science Across Parkinson's (ASAP) Collaborative Research Network, Chevy Chase, Maryland, United States of America

\* j.taanman@ucl.ac.uk

## Abstract

Complex V of the mitochondrial oxidative phosphorylation system is an ATP synthase that plays a pivotal role in the cell's energy transduction. Mutations in genes encoding the multiple protein subunits that constitute complex V cause severe metabolic and neurodegenerative diseases. We present here three complementary assays to assess Complex V activity and assembly in peripheral blood mononuclear cells (PBMCs). The assays involve spectrophotometric and in-gel activity measurements, cytochemical assessment of the mitochondrial transmembrane electrochemical gradient ($\Delta\Psi_m$) to determine if the enzyme acts forward as an ATP synthase or in reverse as an ATPase, and western blot analysis of clear native gels to evaluate Complex V assembly. The whole process can be performed with $2 \times 10^6$ PBMCs isolated from ~2 ml of blood. Our study suggests that PBMCs can serve as a platform for small-scale, minimally invasive investigations of patients suspected of Complex V deficiency or in biomarker research of mitochondrial function.

## Introduction

The oxidative phosphorylation system consists of five multi-subunit enzyme complexes located in the mitochondrial cristae membranes [1]. Complex V (CX-V), also known as ATP synthase, produces the majority of ATP in the cell through phosphorylation of ADP, driven by the transmembrane proton electrochemical gradient generated by the respiratory chain composed of Complexes I to IV [2]. In addition, CX-V plays a key role in the structural formation of the cristae membranes by inducing membrane curvature [3]. The enzyme consists of two functional domains: $F_1$, positioned in the mitochondrial matrix, and $F_o$, embedded in the cristae membrane. A flow of protons from the mitochondrial intermembrane space to the matrix through $F_o$ induces conformational changes in $F_1$ that catalyze the phosphorylation of ADP [2,4,5]. When the transmembrane proton electrochemical gradient is low, CX-V can

**Data availability statement:** All relevant data are within the manuscript and its Supporting Information files.

**Funding:** AHVS, Aligning Science Across Parkinson's (grant number ASAP-000420) through The Michael J. Fox Foundation for Parkinson's Research, https://www.parkinsonsroadmap.org JWT, Fund 42 of the Royal Free Charity, https://www.royalfreecharity.org. The funders had no role in study design, data collection and analysis, decision to publish, or preparation of the manuscript.

**Competing interests:** The authors have declared that no competing interests exist.

work in reverse as an ATPase, hydrolyzing ATP to ADP and using the free energy of this reaction to pump protons from the matrix to the intermembrane space, thus restoring the transmembrane proton electrochemical gradient [6].

Oligomycin A specifically inhibits the proton transfer function of $F_o$ (hence its subscript). The lypophilic cation tetramethylrhodamine methyl ester (TMRM) is a red fluorescent dye that is imported into the mitochondrial matrix dependent on the transmembrane electrochemical gradient ($\Delta\Psi_m$) component of the transmembrane proton electrochemical gradient. It is commonly used as a reporter for $\Delta\Psi_m$ [7]. Addition of oligomycin A to a cell culture will result in $\Delta\Psi_m$ hyperpolarization (increase in TMRM staining intensity) when, prior to its addition, the respiratory chain was functional and CX-V operated forward as an ATP synthase. Conversely, addition of oligomycin A will result in $\Delta\Psi_m$ depolarization (decrease in TMRM staining intensity) when, prior to its addition, the respiratory chain was dysfunctional and CX-V operated in reverse as an ATPase [8].

Human CX-V is comprised of 18 protein subunits [2]. Two subunits are encoded on the mitochondrial DNA (MTATP6 and MTATP8), while the remaining subunits are encoded on the nuclear DNA. Moreover, several nuclear-encoded assembly factors, including ATPAF1, ATPAF2, FMC1, TMEM70 and TMEM242, are required to build a functional CX-V [9]. Mutations in genes involved in CX-V biosynthesis cause CX-V deficiency. They are associated with severe metabolic defects and neurodegenerative disorders, ranging from neonatal onset Leigh syndrome and encephalocardiomyopathy to early- or late-onset neuropathy, ataxia and retinitis pigmentosa [10]. The first genetic mutations causing isolated CX-V deficiency were identified in the mitochondrial *MTATP6* gene coding for a subunit of $F_o$ in the early 1990s [11,12]. More recently, mutations in several nuclear genes (*ATP5F1E*, *ATPAF2* and *TMEM70*) involved in CX-V biosynthesis have been identified [13–15] and since the advent of next-generation sequencing their number is rapidly increasing (reviewed in [9]). In addition, a decline in CX-V activity and integrity with age has been reported in rodent models [16–18].

Because CX-V deficiency is increasingly recognized as a cause of disease, there is a need for a small-scale, minimally invasive CX-V activity assay to evaluate patients. The currently available spectrophotometric CX-V assay is generally performed with isolated mitochondria because the measurement is not sensitive enough to use whole cell extracts [19]. In addition, in-gel activity staining of CX-V with lead(II) nitrate has been reported using whole cell extracts [20] but we found that the white lead(II) phosphate precipitate is challenging to quantify. Both assays measure the reverse ATPase activity of detergent-solubilized CX-V; they do not reveal if CX-V acts forward as an ATP synthase or in reverse as an ATPase in intact mitochondria. Whether CX-V works forward or in reverse can be determined with isolated mitochondria in an Agilent Seahorse Flux Analyzer [21]. However, isolation of mitochondria for the documented spectrophotometric and Seahorse analyses will require a relatively large tissue sample. We have developed three small-scale methods that do not involve the isolation of mitochondria to assess the activity and integrity of CX-V from peripheral blood mononuclear cells (PBMCs). Together the assays require as few as $2\times10^6$ PBMCs, which can be isolated from ~2 ml of whole blood.

## Results

### Spectrophotometric PBMC CX-V activity

PBMCs represent an easily accessible primary tissue and is, therefore, attractive to assess mitochondrial function in patients. Typically, $1 \times 10^6$ of PBMCs can be collected from 1 ml of whole blood. However, a substantially larger number of PBMCs is required to conduct oxidative phosphorylation complex activity assays by spectrophotometry, even when the measurements are adapted to a mini-assay format in 96-well plates or 'dipstick' assays. Therefore, we modified the CX-V activity assay of the commercially available Mitotox Complex V OXPHOS Activity Assay Kit (Abcam) designed to test the inhibitory effect of compounds on CX-V activity. The kit is based on a method first described by Soper and Pedersen [22], where the reverse CX-V (ATPase) activity is determined by an enzyme-linked system that couples the production of ADP to the oxidation of NADH to NAD⁺, which can be monitored as a decrease in optical density at 340 nm ($OD_{340nm}$) with a spectrophotometer or microplate reader.

First experiments were conducted with bovine heart mitochondria as source of CX-V. In time course measurements, 2 µg of bovine heart mitochondria showed a very strong, linear decrease in $OD_{340nm}$. Addition of the CX-V specific inhibitor oligomycin A to bovine heart mitochondria resulted in a much weaker linear decrease in $OD_{340nm}$, whereas assays without bovine heart mitochondria or with only oligomycin A showed no change in $OD_{340nm}$ (Fig 1A). These experiments confirm that bovine heart mitochondria are a rich source of ATPase activity as there is a clear decrease in $OD_{340nm}$ over time. Most of this ATPase activity stems from the reverse CX-V activity, which is oligomycin A sensitive. However, there appears to be also other ATPases active in bovine heart mitochondria because there is still a small decrease in $OD_{340nm}$ in the presence of oligomycin A. The difference between the decrease in $OD_{340nm}$ of bovine heart mitochondria without oligomycin A and the decrease in $OD_{340nm}$ of bovine heart mitochondria with oligomycin A represents the reverse CX-V activity (i.e., the oligomycin A-sensitive ATPase activity). In our experiment, 90.3% of the total bovine heart mitochondrial ATPase activity was oligomycin A-sensitive and is assumed to originate from the reverse CX-V activity. Replication experiments indicated that the intra- and inter-assay coefficients of variation were 3% and 11%, respectively. The experiments were performed with oligomycin A at a final concentration of 62.5 nM because in preliminary titration experiments this oligomycin A

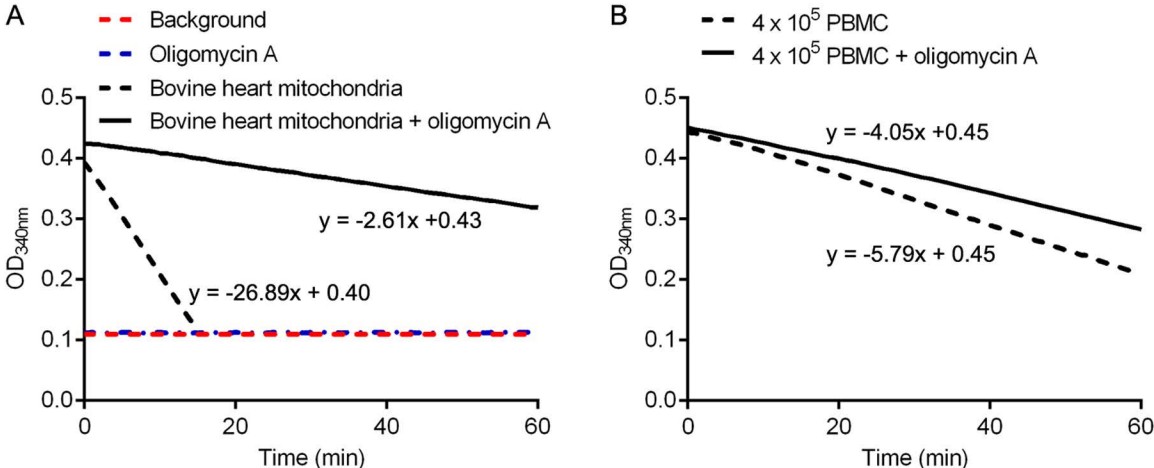

**Fig 1. The decrease in $OD_{320nm}$ over time in spectrophotometric ATPase activity assays. (A)** Assays of 2 µg of bovine heart mitochondria with and without 62.5 nM oligomycin A, and assays without bovine heart mitochondria (background) or with only 62.5 nM oligomycin A. **(B)** Assays of $4 \times 10^5$ PMBCs with or without 62.5 nM oligomycin A. The equations of the trendlines are indicated. The slopes of the trendlines ($\Delta OD_{320nm}$/min) were used to calculate the oligomycin A-sensitive, reverse CX-V activity relative to the total ATPase activity.

concentration gave the maximum inhibition of measured ATPase activity (S1 Fig). Apparently, lower oligomycin A concentrations are not sufficient to fully inhibit CX-V, whereas higher oligomycin A concentrations cause off-target effects.

In following experiments, we found that the kit can be adapted to measure the CX-V activity of a small sample of PBMCs without using the immunoaffinity step of the kit's protocol but, instead, using a low concentration of the detergent digitonin to permeabilize the cells, while not dissociating the oxidative phosphorylation complexes. In titration experiments, we found that 0.01% digitonin resulted in the maximum ATPase activity (S2 Fig). This concentration of digitonin was used in all further experiments. We also tested the effect of PBMC number in the assays. As expected, the ATPase activity was proportional to the number of PMBCs (S3 Fig). Addition of 62.5 nM oligomycin A resulted in a decrease of ATPase activity of 12%, 39%, 39% and 41% when $8 \times 10^5$, $4 \times 10^5$, $2 \times 10^5$ or $1 \times 10^5$ PBMCs were used, respectively (S3 Fig). We chose $4 \times 10^5$ PMBCs as optimal number considering its relatively high total ATPase activity and percentage inhibition by oligomycin A. A typical experiment with two 0.01% digitonin-solubilized samples of $4 \times 10^5$ PBMCs pipetted into two sister wells of a 96-well plate, one without and one with 62.5 nM oligomycin A, is shown in Fig 1B. The sample without oligomycin A showed a clear linear decrease in $OD_{340nm}$. With oligomycin A, the decrease in $OD_{340nm}$ was less. The difference between the slopes indicated that the (oligomycin A sensitive) reverse CX-V activity was 30% of the total ATPase activity. Replication experiments showed that the intra- and inter-assay coefficients of variation were 2% and 7%, respectively. Importantly, no noticeable difference in activity was observed between fresh or previously frozen PBMC samples. Thus, PBMCs can be isolated and stored at -70°C for assays at a later date.

We validated our assay with a CX-V inhibition model of PBMCs, in which the cells were cultured in the absence of oligomycin A or in the presence of increasing concentrations of oligomycin A (1, 10 and 100 nM) for 3 d prior to measurement of CX-V activity. We hypothesized that long-term exposure of PBMCs in culture to a low concentration of a stressor like oligomycin A would affect the integrity and activity of CX-V. As shown in Fig 2, exposure to oligomycin A as stressor in culture caused a statistically significant dose-dependent decline in reverse CX-V activity relative to total ATPase activity, from on average ~28% in untreated cells (0 nM oligomycin) to on average ~6% in 100 nM oligomycin A-treated cells. Thus, our hypothesis that long-term exposure to oligomycin A in culture affects CX-V activity appears correct.

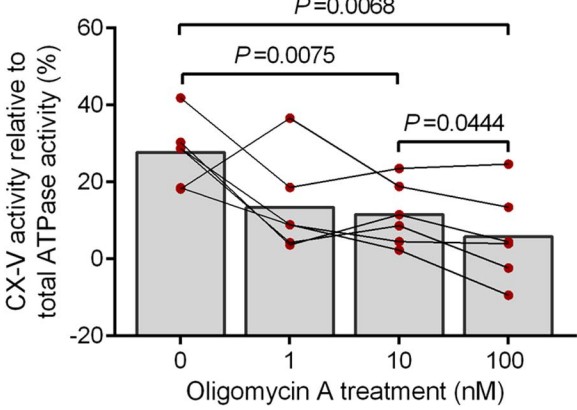

**Fig 2. Reverse CX-V activity in PBMCs cultured in the presence of increasing concentrations (0–100 nM) of oligomycin A as stressor for 3 days.** Shown is the mean percentage of the reverse CX-V activity relative to total ATPase activity in PBMCs cultured for 3 d in the absence of oligomycin A (0 nM) or in the presence of increasing concentrations of oligomycin A (1, 10 and 100 nM) as stressor (n=6). After culturing, the cells were assayed without and with 62.5 nM oligomycin A to determine the total and oligomycin A sensitive ATPase activity, respectively. The oligomycin A sensitive ATPase activity represents the reverse CX-V activity. Individual data points of the six cultures are shown; data points from corresponding cultures are connected. Statistically significant differences are indicated with their P values.

## Analysis of ΔΨ$_m$

Incubation of cell cultures with 25 nM TMRM produces specific mitochondrial fluorescent staining proportional to ΔΨ$_m$ [7,8]. To investigate the ΔΨ$_m$ of the PMBCs cultured in the presence of 0–100 nM oligomycin A as stressor for 3 d, cells were subsequently stained with TMRM, followed by flow cytometric analysis. To determine whether CX-V works forward as an ATP synthase or in reverse as an ATPase, the TMRM staining was carried out without or with oligomycin A, rotenone+antimycin A, or FCCP. We first investigated cells cultured for 3 d without oligomycin A as stressor (0 nM). Addition of oligomycin A during the TMRM staining of PBMCs resulted on average in a ~48% increase of gated cells (i.e., cells with a high TMRM staining intensity) compared to cells stained in absence of oligomycin A (Fig 3). This finding is in line with the expectation that addition of oligomycin A during the staining will result in ΔΨ$_m$ hyperpolarization when, prior to oligomycin A addition, the respiratory chain is functional and CX-V acts forward as an ATP synthase [7,8]. Next, the CX-I inhibitor rotenone and the CX-III inhibitor antimycin A were added during the TMRM staining of PBMCs cultured for 3 d without oligomycin A as stressor. This resulted on average in a ~36% decrease in gated cells compared to cells stained with TMRM in absence of rotenone and antimycin A (Fig 3). This is consistent with the notion that ΔΨ$_m$, maintained by the respiratory chain, will decrease when respiratory chain activity is inhibited and CX-V functions in reverse as an ATPase [7,8]. We also added the protonophore FCCP during the TMRM staining of PBMCs cultured for 3 d without oligomycin A as stressor and found on average a ~52% decrease in gated cells compared to cells stained with TMRM in absence of FCCP (Fig 3). This observation confirms that the unrestricted transfer of protons across the membrane by the protonophore causes ΔΨ$_m$ depolarization.

When we investigated the PBMCs cultured for 3 d with increasing concentrations (1–100 nM) of oligomycin A as stressor, we noticed a statistically significant decrease in the fraction of gated cells analyzed with TMRM in the presence of oligomycin A compared to cells analyzed in the absence of oligomycin A (Fig 3). Apparently, ΔΨ$_m$ decreases during long-term exposure of oligomycin A as stressor in culture. Addition of rotenone+antimycin A or FCCP during TMRM analysis of PBMCs cultured with increasing concentrations of oligomycin A as stressor showed no clear differences in the fraction of gated cells compared to PBMCs cultured in the absence of oligomycin A (Fig 3), suggesting that these toxins do not further affect ΔΨ$_m$ during long-term exposure of oligomycin A in culture.

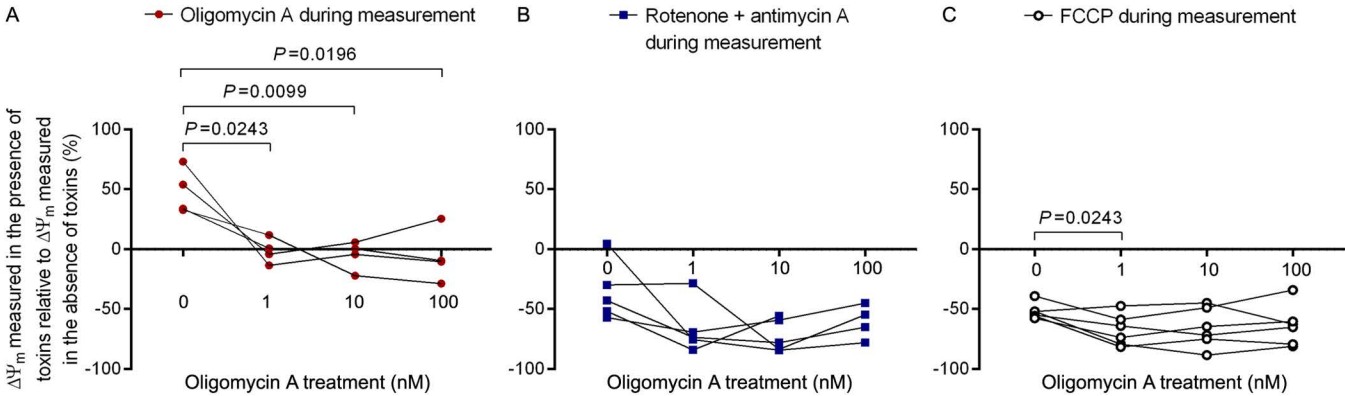

**Fig 3. ΔΨ$_m$ of PMBCs cultured for 3 d in the presence of increasing concentrations (0–100 nM) of oligomycin A as stressor and subsequently analyzed in the absence or presence of the mitochondrial toxins oligomycin A, rotenone +antimycin A or FCCP.** Shown are the individual data points of the ratio (%) of ΔΨ$_m$ from cultures analyzed (A) with and without oligomycin A (n = 4), (B) with and without rotenone+antimycin A (n = 5), or (C) with and without FCCP (n = 6). Data points from corresponding cultures are connected. Statistically significant differences are indicated with their *P* values.

## In-gel PBMC CX-V activity and assembly

In addition to spectrophotometric assays and cytochemical staining procedures, blue native polyacrylamide gel electrophoresis [23] has been used to study the activity and integrity of CX-V. Combined with western blot analysis, blue native gels can be exploited to study the assembly and stability of CX-V [24–26], whereas in combination with a histochemical staining procedure CX-V activity can be revealed [27,28]. However, instead of blue native gels, we used clear native gels to investigate CX-V of PBMCs. Clear native gels are a variant of the more frequently used blue native gels [29,30]. The advantage of the colorless clear native gels is that they allow better detection of the white lead(II) phosphate precipitate detection of ATPase activity without interference of the remnant Coomassie Blue staining of blue native gels. Nevertheless, we found the white lead(II) phosphate precipitate problematic to visualize. For this reason, we enhanced the in-gel CX-V staining by converting white lead(II) phosphate to black lead(II) sulfide (S4 Fig) [31], which increased the sensitivity of the in-gel CX-V staining dramatically.

To determine the detection limit for CX-V in-gel activity, we resolved a serial dilution of extracted PMBCs on a clear native gel, followed by ATPase staining. The staining procedure revealed a single band migrating at ~700 kDa (Fig 4A), which corresponds to the migration of holo-CX-V. Therefore, we conclude that this band represents reverse CX-V activity. The staining intensity was proportional to the number of PBMCs (Fig 4B). Extracts derived from between ~2 × 10⁵ to ~8 × 10⁵ PBMCs showed well-defined staining; the detection limit was ~1 × 10⁵ PBMCs. Extracts derived from ≥9 × 10⁵ PBMCs resulted in overloading (i.e., a distortion of the band; S5 Fig).

In the next experiment, we used -70°C stored samples of the cultured PBMCs exposed to 0–100 nM oligomycin A as stressor for 3 d. Similar to our findings shown in Fig 2, long-term oligomycin A-exposure in culture produced a statistically significant, dose-dependent decrease in CX-V activity (Fig 5A, 5B). A western blot of a clear native gel, ran in parallel with the stained gel and probed with an antibody raised against the ATP5A subunit of CX-V, showed a statistically significant, dose-dependent decrease of assembled CX-V comparable to the decrease in CX-V activity staining (Fig 5A, 5C).

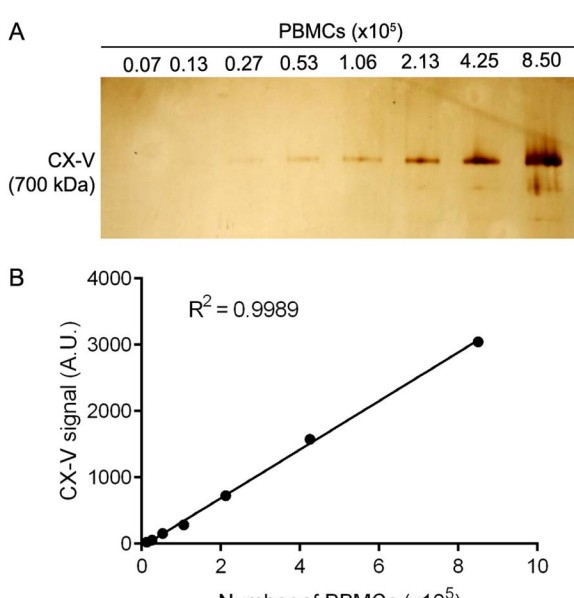

**Fig 4. Determination of the detection limit of in-gel CX-V activity staining. (A)** In-gel CX-V activity staining of extracts from increasing numbers of PBMCs. **(B)** Quantification of the in-gel CX-V activity signals in extracts from increasing numbers of PBMCs.

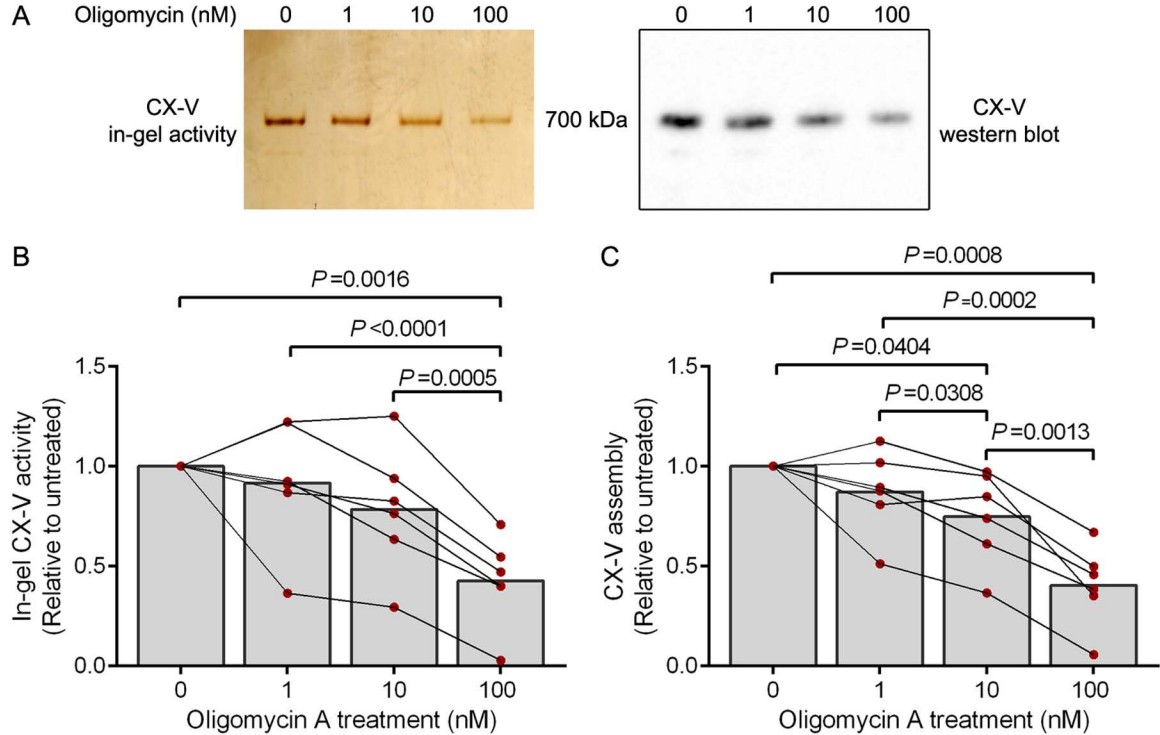

**Fig 5. In-gel CX-V activity and assembled CX-V in PBMCs cultured in the presence of increasing concentrations of oligomycin A for 3 days.**
**(A)** In-gel CX-V activity staining ($2.5 \times 10^5$ cells) and western blot ($2.5 \times 10^4$ cells) of clear native gels of a representative PBMC culture probed with an antibody against the ATP5A subunit of CX-V. **(B)** Quantification of the in-gel CX-V activity signals. **(C)** Quantification of the assembled CX-V signals on the western blots. Shown is the mean in-gel CX-V activity signal or CX-V assembly signal relative to the signal in cultures not treated for 3 d with oligomycin A (0 nM; n = 6). Individual data points of the six cultures are shown; data points from corresponding cultures are connected. Statistically significant differences are indicated with their *P* values.

Apparently, long-term exposure to oligomycin A in culture affects the stability of CX-V and this results in a decrease of CX-V activity.

To investigate the diagnostic potential of the in-gel CX-V activity stain, we compared cultured dermal fibroblasts from four control subjects with those from a patient harboring an *MRPS25* gene mutation, as we currently do not have PMBC samples from patients with a CX-V deficiency. *MRPS25* encodes a protein constituent of the 28S small subunit of the mitochondrial ribosome. The mutation causes a mitochondrial translation defect [32], which is expected to impair CX-V biogenesis as two of the 18 CX-V subunits are encoded on the mitochondrial DNA. Fibroblast extracts from the four control subjects showed robust in-gel CX-V activity staining, whereas the fibroblast extract from the patient showed a weak band, indicating a CX-V deficiency (S6 Fig). This result suggests that the in-gel CX-V staining is likely to expose CX-V deficiency in PBMC samples.

## Discussion

We developed three complementary assays to assess the activity and integrity of CX-V from PBMCs. The assessment can be performed with $2 \times 10^6$ PBMCs isolated from as little as 2 ml of blood. Our work suggests that PBMCs can be exploited for minimally invasive detection of CX-V deficiency in patients or in biomarker studies of mitochondrial function. Although the analysis of $\Delta\Psi_m$ needs to be carried out with freshly isolated PBMCs, the other assays can conveniently be performed after storage at -70˚C. If $< 2 \times 10^6$ PBMCs are available, the gel stained for CX-V activity can be electro-blotted

after staining to prepare a western blot rather than using a companion gel. We found, however, that this decreases the sensitivity of the CX-V immuno-detection.

To validate the assays, we cultured PBMCs for 3 d in the presence of 0–100 nM oligomycin A. We assumed that 3-d exposure of PBMCs in culture to oligomycin A as a stressor would cause CX-V deficiency. As expected, the spectrophotometric activity assay revealed an oligomycin A dose-dependent decrease in CX-V activity. This finding was confirmed in the in-gel activity assays. Western blot analysis of corresponding clear native gels showed that the CX-V assembly levels decreased in an oligomycin A dose-dependent manner. Thus, it appears that long-term exposure of PBMCs to oligomycin A in culture affects the stability of holo-CX-V, causing a decrease in CX-V activity. To detect assembled CX-V on the western blot, we used an antibody against the ATP5A subunit of the $F_1$ portion of CX-V. Potential assembly defects of CX-V can be further scrutinized with antibodies against other subunits of CX-V. We prepared our own polyacrylamide gradient gels for clear native gel electrophoresis. However, the clear native in-gel staining and western analysis can also be performed with commercially available (blue) native polyacrylamide gels when the anode and cathode buffers and sample preparation are adapted as outlined in the methods section.

Both the spectrophotometric and the in-gel CX-V activity assay measure the reverse CX-V (ATPase) activity in PBMC samples disrupted with the mild detergents digitonin or n-dodecyl-β-D-maltoside, respectively. In contrast, the analysis of $\Delta\Psi_m$ is carried out with intact cells. We found that addition of oligomycin A during the TMRM-staining to inhibit CX-V led to an increase in $\Delta\Psi_m$ compared to cells not treated with oligomycin A during the staining. This indicates that CX-V operates forward as an ATP synthase and that PBMCs have a functional respiratory chain in undisrupted PBMCs [8]. Remarkably, after long-term exposure to oligomycin A as a stressor in culture, addition of oligomycin A during the TMRM-staining to inhibit CX-V caused a decrease in $\Delta\Psi_m$ compared to cells not treated with oligomycin A during the staining. This indicates that long-term exposure to oligomycin A as a stressor in culture impairs respiratory chain function and CX-V acts in reverse as an ATPase [8]. The western blot analysis showed that long-term exposure to oligomycin A as a stressor in culture leads to a decline in assembled CX-V. In addition to its catalytic function, CX-V plays a vital role in the structural formation of the cristae membranes where the respiratory chain enzyme complexes are located [3]. Low levels of CX-V after long-term exposure to oligomycin are, therefore, likely to lead to disruption of the cristae membranes, which might explain the respiratory chain enzyme deficiency.

Replication experiments of the spectrophotometric assays with the same PBMC sample indicated relatively small intra- and inter-assay coefficients of variation. In the experiments in which cultured PBMCs were exposed to oligomycin A as a stressor, four to six PBMC samples were tested from different subjects. Although all samples showed a similar qualitative response to increasing oligomycin A exposure in culture, the quantitative data varied noticeably between the samples. We assume that the quantitative differences are caused the natural variation of the samples.

## Conclusion

We have developed three small-scale assays to evaluate CX-V activity and assembly in PBMCs. The assays include spectrophotometric and in-gel activity measurements, cytochemical assessment of $\Delta\Psi_m$ and western blot analysis of clear native gels to investigate CX-V assembly. The assays can be carried out with $2 \times 10^6$ PBMCs isolated from 2 ml of whole blood. Our investigations suggest that PBMCs can serve as a platform for minimally invasive investigations of patients suspected of CX-V deficiency.

## Materials and methods

### Stock solutions

The following stock solutions were prepared: antimycin A (Merck), 10 mM in dimethylsulfoxide (DMSO); carbonyl cyanide-p-trifluoromethoxyphenylhydrazone (FCCP; Merck), 10 mM in DMSO; digitonin (Merck), 5% in water; glycerol (Merck), 50% in water; leupeptin (Merck), 1 mg/ml in water; n-dodecyl-β-D-maltoside (Thermo Fisher Scientific), 20% in

water; oligomycin A (Merck), 10 mM in DMSO; pepstatin A (Merck), 1 mg/ml in methanol; phenylmethylsulphonyl fluoride (PMSF; Merck), 1 M in acetone; Ponceau S (Merck), 0.1% in water; rotenone (Merck), 10 mM in DMSO; TMRM (Thermo Fisher Scientific), 25 µM in DMSO. Stock solutions of antimycin A, FCCP, leupeptin, oligomycin A, PMSF, pepstatin A, rotenone and TMRM were stored at -20˚C, stock solutions of n-dodecyl- β-D-maltoside and Ponceau S were stored at 4˚C, while stock solutions of digitonin and glycerol were stored at room temperature.

## PBMC isolation, culturing and oligomycin A treatment

Ethical approval was obtained from the UCL/UCLH Biobank for Studying Health and Disease – New Collection NC01.21. All donors gave prior informed, written consent and all work was carried out in compliance with the Declaration of Helsinki and national legislation. The dates when blood samples were accessed for research purposes were: 01/07/2024, 22/08/2024 and 17/02/2025. PBMCs were isolated from blood donated by healthy adults (S1 Table) using the density gradient centrifugation procedure with Lymphoprep (Stemcell Technologies) [33]. After counting [33], PBMC suspensions were used fresh or stored at -70˚C as indicated below.

Fresh PBMCs were cultured at a density of $10^6$ cells per ml in RPMI medium 1640 with Glutamax and 25 mM HEPES (Thermo Fisher Scientific), supplemented with 10% fetal bovine serum (FBS; Thermo Fisher Scientific) and 1 × Antibiotic Antimycotic Solution (Merck) in a humidified atmosphere at 5% $CO_2$, 37˚C. Oligomycin A to final specified concentrations was added as stressor when the cultures were set up and after 24 h. After 3 d, cells were collected and washed in phosphate-buffered saline (PBS, Thermo Fisher Scientific).

Primary skin fibroblasts from four control subjects and a patient with a homozygous c.215C > T mutation in *MRSP25* [32] were cultured in DMEM with Glutamax (Thermo Fisher Scientific), supplemented with 10% FBS, 1 mM sodium pyruvate (Merck), 50 units/ml of penicillin (Thermo Fisher Scientific), 50 µg/ml of streptomycin (Thermo Fisher Scientific) and 200 µM uridine (Merck) as described [33]. For harvesting, cells were dislodged by trypsinization, collected by centrifugation and washed once in PBS. Cell pellets were stored at -70°C.

## Spectrophotometric measurement of CX-V activity

CX-V activity was measured with the Mitotox Complex V OXPHOS Activity Assay Kit supplied by Abcam, using a modified protocol. Initial experiments with bovine heart mitochondria as source of CX-V (provided by the kit) were performed as recommended by the manufacturer. In further experiments, PBMC suspensions were used. A fresh or previously frozen PMBC suspension was counted [33] and two samples of $4 \times 10^5$ PBMCs were pelleted by centrifugation at $17,000 \times g$ for 10 min (see S7 Fig for workflow). Each pellet was resuspended in 4 µl of PBS and loaded into two sister wells of a 96-well plate, followed by 40 µl of Phospholipids provided by the kit and digitonin stock solution to final concentration of 0.01%. Oligomycin A stock solution was added to one of the sister wells to a final concentration of 62.5 nM to specifically inhibit CX-V. The plate was incubated at room temperature for 45 min. Then, 200 µl of Complex V Activity Buffer provided by the kit was added to each well, supplemented with digitonin and with or without oligomycin A at the same final concentrations. Lastly, the optical density at 340 nM ($OD_{340nm}$) was measured every minute for 60 min using a BioTek Cytation-1 (Agilent) microplate reader. The $OD_{340nm}$ of both wells were plotted against time and the linear range of the decrease in $OD_{340nm}$ ($\Delta OD_{340nm}$) was identified. To calculate the (oligomycin A-sensitive) CX-V activity as a percentage relative to the total ATPase activity we used the following formula: CX-V activity = $(A-B)/A \times 100\%$, where $A$ is the $\Delta OD_{340nm}$/min (i.e., slope of the linear range) of the well without oligomycin and $B$ is the $\Delta OD_{340nm}$/min of the well with oligomycin.

## TMRM staining

A fresh PBMC suspension was counted [33] and samples of ≥ $4.0 \times 10^4$ PBMCs were pelleted at $17,000 \times g$ for 10 min (see S7 Fig for workflow). Pellets were resuspended in 50 µl of PBS containing 25 nM TMRM, followed by incubation at 37˚C for 15 min. Then, another 50 µl of PBS containing 25 nM TMRM, with or without oligomycin A (1.5 µM final), or FCCP (1.5

µM final), or rotenone+antimycin A (both 1 µM final) were added and further incubated at 37°C for 15 min. Next, 70 µl of the cell suspension, representing typically $3.0 \times 10^4$ PBMCs, were analyzed with a Moxi GO II cell analyzer fitted with a 561 nm LP filter (Orflo Technologies). A gate was applied to cells >6 µm and 2 log of the fluorescence to determine the fraction of TMRM-positive (gated) cells against the total population of cells. This fraction was obtained from cells analyzed in the absence or presence of oligomycin A, FCCP or rotenone+antimycin A. Then the fraction of the TMRM-positive cells analyzed in the presence of oligomycin A, FCCP or rotenone+ antimycin A was expressed relative to the fraction of the TMRM-positive cells analyzed in the absence of these toxins to obtain the change in $\Delta\Psi_m$.

## Clear native gel electrophoresis, in-gel CX-V activity staining and western blot analysis

All reagents were purchased from Merck, except that n-dodecyl-β-D-maltoside was from Thermo Fisher Scientific, anti-ATP5A mouse monoclonal antibody (clone 15H4C4) from Abcam and anti-mouse IgG HRP conjugate from Promega.

Previously frozen PBMC suspensions with a known cell number were pelleted by centrifugation at $17,000 \times g$, 4°C for 10 min. Cell pellets were resuspended in 1 M 6-aminocaproic acid, 50 mM bistris, 0.5% n-dodecyl-β-D-maltoside, 1 mM PMSF, 1 µg/ml of leupeptin and 1 µg/ml of pepstatin A (pH 7.0) on ice for 15 min, followed by 15-min centrifugation at $17,000 \times g$, 4°C and collection of the supernatants [34]. The 50% glycerol stock solution was added to the cell extracts to a final concentration of 5% and the 0.1% Ponceau S stock solution was added to a final concentration of 0.01% [29]. Clear native gel electrophoresis was performed with the Mini Protean 3 System (BioRad Laboratories). Samples were loaded onto native 3–12% gradient polyacrylamide gels with a 3% stacking gel prepared as described [35]. As anode buffer was used 50 mM bistris (pH 7.0) and as cathode buffer was used 50 mM tricine, 15 mM bistris, 0.05% Triton X-100 and 0.05% sodium deoxycholate (pH 7.0) [29]. Gels were run at 100 V constant for 15 min, followed by 4 mA constant per gel until the Ponceau S dye ran off.

For in-gel CX-V activity staining, gels were rinsed with water, followed by a 2-h incubation in 50 ml of 34 mM Tris, 270 mM glycine, 14 mM $MgSO_4$, 6 mM $Pb(NO_3)_2$ and 8 mM ATP (pH 7.8) at 37°C, while gently shaking [27]. Gels were rinsed twice with water, prior to conversion of the white $Pb_3(PO_4)_2$ precipitate stain to a black PbS precipitate stain through a brief (5–10 s) rinse with 1.0% $(NH_4)_2S$ solution in a hood (S4 Fig) [31]. The reaction was stopped by several rinses with water. Gels were photographed on a light box and bands were quantified with ImageJ [36].

For western blot analysis, gels were electro-blotted onto Immun-Blot PVDF Membrane (Biorad Laboratories) as described [37], followed by 1-h blocking in 10% skimmed milk powder, PBS at room temperature. Blots were incubated with a $10,000^{-1}$ diluted mouse monoclonal primary antibody against ATP5A in 0.3% Tween-20, PBS at 4°C, overnight and then further washed, incubated with a secondary antibody ($6,000^{-1}$ diluted anti-mouse IgG HRP conjugate), washed again, developed and imaged as described [38].

## Statistical analyses

Graphs and statistical analyses were executed with GraphPad Prism software. Data were assessed for normal distribution using the Shapiro-Wilk test. Statistical significance was analysed using one-way ANOVA with Greenhouse-Greisser correction, followed by Tukey's multiple comparison test. Statistical significance levels were set to $P < 0.05$.

## Supporting information

**S1 Table. Age and sex of PBMC donors.**
(PDF)

**S1 Fig. Effect of increasing oligomycin A concentrations on the decrease in $OD_{320nm}$ ($\Delta OD_{320nm}$) per minute in spectrophotometric ATPase activity assays of PBMCs.** Measurements of $4 \times 10^5$ PBMCs solubilized with 0.01% of digitonin. Two separate experiments are shown.
(PDF)

**S2 Fig. Effect of increasing digitonin concentrations on the decrease in $OD_{320nm}$ ($\Delta OD_{320nm}$) per minute in spectrophotometric ATPase activity assays of PBMCs.** Measurements of $4 \times 10^5$ PBMCs without oligomycin A. Two separate experiments are shown.
(PDF)

**S3 Fig. Effect of increasing PBMC cell numbers on the decrease in $OD_{320nm}$ ($\Delta OD_{320nm}$) per minute in spectrophotometric ATPase activity assays measured without or with 62.5 nM oligomycin A.** PBMCs were solubilized with 0.01% of digitonin.
(PDF)

**S4 Fig. Chemical reactions of the in-gel CX-V staining procedure.** The $F_1$ and $F_o$ moieties of CX-V are outlined. $P_i$, inorganic orthophosphate.
(PDF)

**S5 Fig. Determination of the detection limit of in-gel CX-V activity staining.** (A) In-gel CX-V activity staining of extracts from increasing numbers of PBMCs. (B) Quantification of the in-gel CX-V activity signals in extracts from increasing numbers of PBMCs. Note the distortion of the band when overloaded with an extract derived from $9 \times 10^5$ PBMCs.
(PDF)

**S6 Fig. In-gel CX-V activity of 15-μg protein extracts from cultured fibroblast of four control subject and one patient (P) with a mitochondrial translation deficiency.**
(PDF)

**S7 Fig. Workflow of the spectrophotometric CX-V assay and $\Delta\Psi_m$ analysis.**
(PDF)

**S1 File. Data and calculations for Figs 1, 2, 3, 4B, 5B and 5C, and S1 Fig, S2 Fig, S3 Fig and S5 Fig.**
(XLSX)

**S1 Raw images. Flow cytometry graphs upon TMRM staining.** Left panel: raw scans upon different treatments. A gate was applied to cells >6 μm (X-axis, linear scale) and 2 log fluorescence (Y-axis, log scale). Right panel: Comparisons were made between untreated PBMCs (UT, grey background) and PBMCs with added oligomycin A (blue curves), rotenone+antimycin A (red curves) or FCCP (pink curves) drawn in a histogram (fluorescence versus cell number).
(PDF)

**S2 Raw images. Uncropped, unadjusted images of in-gel activity staining and western blots.**
(PDF)

## Acknowledgments

We would like to thank the blood donors. We would also like to acknowledge Drs Jane Macnaughtan and Marco Toffoli for the collection of blood samples.

## Author contributions

**Conceptualization:** Kai-Yin Chau, Jan-Willem Taanman.

**Data curation:** Kai-Yin Chau, Jan-Willem Taanman.

**Formal analysis:** Kai-Yin Chau, Jan-Willem Taanman.

**Funding acquisition:** Jan-Willem Taanman, Anthony H.V. Schapira.

**Investigation:** Kai-Yin Chau, Jan-Willem Taanman.

**Methodology:** Kai-Yin Chau, Jan-Willem Taanman.

**Resources:** Anthony H.V. Schapira.

**Supervision:** Anthony H.V. Schapira.

**Visualization:** Kai-Yin Chau, Jan-Willem Taanman.

**Writing – original draft:** Kai-Yin Chau, Jan-Willem Taanman.

**Writing – review & editing:** Anthony H.V. Schapira.

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
