## [Decision Letter · Decision Letter 0]

3 Feb 2025

PONE-D-25-01588Small-scale protocols to characterize mitochondrial Complex V activity and assembly in peripheral blood mononuclear cellsPLOS ONE

Dear Dr. Taanman,

Thank you for submitting your manuscript to PLOS ONE. After careful consideration, we feel that it has merit but does not fully meet PLOS ONE’s publication criteria as it currently stands. Therefore, we invite you to submit a revised version of the manuscript that addresses the points raised during the review process.

 Please respond to reviewers' comments individually.

We look forward to receiving your revised manuscript.

Kind regards,

Xiaosheng Tan

Academic Editor

PLOS ONE

2. Thank you for stating the following financial disclosure: [AHVS, ASAP-000420, Aligning Science Across Parkinson’s through The Michael J. Fox Foundation for Parkinson’s Research, https://www.royalfreecharity.org, No

JWT, Fund 42, Royal Free Charity, https://www.royalfreecharity.org, No]. 

Additional Editor Comments (if provided):

Reviewers' comments:

Reviewer's Responses to Questions

**Comments to the Author**

1. Is the manuscript technically sound, and do the data support the conclusions?

Reviewer #1: Yes

Reviewer #2: Yes

Reviewer #3: Yes

2. Has the statistical analysis been performed appropriately and rigorously? 

Reviewer #1: Yes

Reviewer #2: Yes

Reviewer #3: Yes

3. Have the authors made all data underlying the findings in their manuscript fully available?

Reviewer #1: Yes

Reviewer #2: Yes

Reviewer #3: Yes

4. Is the manuscript presented in an intelligible fashion and written in standard English?

Reviewer #1: Yes

Reviewer #2: Yes

Reviewer #3: Yes

5. Review Comments to the Author

Reviewer #1: Chau et al. developed small-scale protocols to characterize mitochondrial Complex V activity and assembly in peripheral blood mononuclear cells (PBMCs). While the study offers valuable insights, several critical areas require further exploration to strengthen the findings and enhance their clinical relevance. Below are detailed suggestions for improvement:

Major Comments:

1. It is strongly recommended to test the assays on PBMCs from patients with known Complex V mutations or mitochondrial diseases. This will validate the sensitivity and specificity of the assays in a clinical setting. Additionally, compare results from patient samples with those of healthy controls to establish baseline variability and assess this assays' diagnostic potential.

2. The authors need to assess Complex V functionality under stress conditions, PBMCs should be treated with metabolic stressors such as glucose deprivation or hypoxia. Furthermore, investigating whether changes in mitochondrial membrane potential (ΔΨm) correlate with ATPase or ATP synthase activity under these conditions will provide deeper insights into the enzyme's adaptability and response to metabolic challenges.

3. Utilizing a Seahorse XF Analyzer is essential to evaluate mitochondrial bioenergetics, including ATP production, basal respiration, and spare respiratory capacity. These analyses will offer a more comprehensive understanding of Complex V functionality in PBMCs.

4. The assays should be tested in other blood-derived cells, such as monocytes, platelets, and lymphocytes, to determine whether they are specific to PBMCs or applicable across different cell types. Additionally, extending testing to non-blood-derived cells, such as fibroblasts, will confirm the versatility of the assays.

5. Measuring reactive oxygen species production and correlating these findings with Complex V dysfunction and ΔΨm changes will provide valuable insights into the broader implications of mitochondrial dysfunction and oxidative stress.

Reviewer #2: The authors developed small-scale assays for evaluating Complex V activity and assembly using as few as 2 million PBMCs, making the process less invasive and feasible with limited blood samples. This is a significant advancement for diagnostic and research applications, especially for diseases involving mitochondrial dysfunction.

This manuscript is easy to read, and the logic is clear and understandable. The experimental approaches are appropriate and provide data to support these findings. However, this work still has a small sample size. Therefore, the reviewer is supportive of its publication until the authors address several aspects as follow:

Major issue

Major issue 1: Limited Sample Size. All the experiments seem to have only 2 replicates (n=2) (there is even a group of data without an error bar in Figure 3). Figure 1, Figure 4, and Figures S1-3 appear to have only n=1. More replicates are needed to strengthen the experimental conclusions. Please ensure that n≥3.

Furthermore, current work did not take into account other factors, such as age and sex, as well as patients with variants in Complex V genes. Including more results with this information would make the authors' small-scale assays more robust and reliable for future potential diagnostic applications.

Major issue 2: Concentration Determination. In the preliminary titration experiment (Figure S1), were these concentrations chosen randomly? Why were more concentrations around 62.5 nM not tested, such as 50-70 nM? The reviewer doubts that another concentration, rather than 62.5 nM, may give the maximum inhibition. A similar question arises regarding Figure S2; why were more concentrations around 0.01% not tested, such as 0.005-0.015%? The reviewer doubts that another digitonin concentration, rather than 0.01%, might result in the maximum ATPase activity.

Major issue 3: The Optimal Number of PBMCs. Lines 117-120 state: 'Addition of 62.5 nM oligomycin A resulted in a decrease of ATPase activity of 12%, 39%, 39%, and 41% when 8 × 10^5, 4 × 10^5, 2 × 10^5, or 1 × 10^5 PBMCs were used, respectively (Figure S3). We chose 4 × 10^5 PBMCs as the optimal number, considering its relatively high total ATPase activity and percentage inhibition by oligomycin A.' The group of 2 × 10^5 PBMCs also gives results very similar to those of the group of 4 × 10^5 PBMCs. Considering that this paper aims to establish a protocol using fewer PBMCs, why not choose 2 × 10^5 PBMCs as the optimal number?

Major issue 4: Discrepancy Between Figures. Figure 5 seems to have an opposite conclusion to Figure S3. In Figure S3, the oligomycin A decrease effect on ATPase activity was attenuated when the PBMC cell number increased. However, Figure 5 shows that a larger PBMC cell number strengthens the oligomycin A decrease effect on ATPase activity. How do the authors explain the discrepancy between the conclusions of these two figures?

Minor issue

Minor issue 1: Lines 59-60, “Human CX-V is comprised of 18 protein subunits. Two subunits are encoded on the mitochondrial DNA, while the remaining subunits are encoded on the nuclear DNA.” Please provide the specific names of the two subunits. Additionally, one of John E. Walker's papers should be cited when discussing the 18 subunits: https://doi.org/10.1073/pnas.1722086115 or https://doi.org/10.1039/9781788010405-00338.

Minor issue 2: Be more specific instead of using general descriptors.

Provide one or two specific names in lines 60-61: “Moreover, several nuclear-encoded assembly factors are required to build a functional CX-V [7].”

Provide one or two specific defect/disorder symptoms or names in lines 62-63: “They are associated with severe metabolic defects and neurodegenerative disorders [8].”

Provide one or two specific gene names in lines 64-66: “More recently, mutations in several nuclear genes involved in CX-V biosynthesis have been identified, and since the advent of next-generation sequencing, their number is rapidly increasing (reviewed in: [7]).”

Provide one or two specific disease names in line 68: “Because CX-V deficiency is increasingly recognized as a cause of disease.”

When specifying these sentences above, please add reference(s) if needed.

Minor issue 3: Line 122 states 'with 62.6 nM oligomycin A.' Is this a typo? Why does it use a different concentration of 62.6 nM, rather than the 62.5 nM mentioned in other contexts?

Minor issue 4: The figure legends for Figures 2 and 3 are ambiguous.

In Figure 2, “the percentage of oligomycin A-sensitive CX-V activity relative to total ATPase activity in samples not treated with 62.5 nM oligomycin A during the measurements (n=2).” How does the total ATPase activity of each group come about? Please specify whether it comes from the PBMCs cultured for 3 days without oligomycin A treatment, or from the fresh PBMCs (cultured for 0 days) without oligomycin A treatment. If it comes from the former, how is the first group's relative value calculated (since it is already the 'control group' without oligomycin A treatment)? In addition to figure S5, please describe the experimental protocol for Figure 2 in more detail in the Methods section. Additionally, it seems unnecessary to mention the 62.5 nM concentration in the figure legend, which is actually 0.

In Figure 3, “the percentage of gated cells treated with the indicated toxins during the measurements compared to the gated cells not treated during the measurements (n=2).” For each test group, does “the gated cells not treated during the measurements” refer to PBMCs cultured for 3 days or fresh PBMCs? Please also describe the experimental protocol for Figure 3 in more detail in the Methods section.

Minor issue 5: Lines 146-148 state, “Addition of oligomycin A during the TMRM staining of PBMCs resulted in a ~33% increase of gated cells (i.e., cells with a high TMRM staining intensity) compared to cells stained in absence of oligomycin A (Fig 3).” Does this sentence refer to the group colored in white? Actually, the white group is treated without oligomycin A in Figure 3.

Minor issue 6: Line 154-155. It would be better to simply describe the FCCP function, in a style like the context in line 150 “the CX-I inhibitor rotenone and the CX-III inhibitor antimycin A.”

Minor issue 7, line 192-193. “Extracts derived from >1× 10^6 PBMCs resulted in overloading (i.e. a distortion of the band).” Please show the related images as a supplemental figure.

Minor issue 8, line 202, “probed with an antibody against a subunit of CX-V”. It’s better to specify the antibody as “antibody against the ATP5A subunit of the F1 portion of CX-V,” instead of only showing the antibody identity in the discussion.

Reviewer #3: This paper aim set up several assays to assess Complex V activity and assembly in peripheral blood mononuclear cells (PBMCs). The study presents a clear and logical progression of ideas, making the conclusions well-supported by the data. The claims made by the authors are supported by strong evidence, reinforcing the validity of their conclusions.

My concerns and suggestions are:

1. You should mention the previous method used to assess Complex V activity and highlight its limitations. Then, discuss your improvements, focusing on enhanced sensitivity, specificity, accuracy, or technical advantages.

2. The color in figure3 need to be changed to make the figure accessible for individuals with red-green color blindness, adjust the color scheme by avoiding red-green combinations.

3. It is kind of unnormal that 6 x105 PMBC has less CX-V in-gel activity than 3 x 105, Can you explan it? Or discussion it in the result?

6. PLOS authors have the option to publish the peer review history of their article (what does this mean? ). If published, this will include your full peer review and any attached files.

**Do you want your identity to be public for this peer review?** For information about this choice, including consent withdrawal, please see our Privacy Policy .

Reviewer #1: **Yes: ** Jing Ju

Reviewer #2: No

Reviewer #3: No

---

## [Author Response · Author response to Decision Letter 1]

19 Mar 2025

Reviewer #1: Chau et al. developed small-scale protocols to characterize mitochondrial Complex V activity and assembly in peripheral blood mononuclear cells (PBMCs). While the study offers valuable insights, several critical areas require further exploration to strengthen the findings and enhance their clinical relevance. Below are detailed suggestions for improvement:

We thank Reviewer 1 for the time and consideration to review our manuscript.

Major Comments:

1. It is strongly recommended to test the assays on PBMCs from patients with known Complex V mutations or mitochondrial diseases. This will validate the sensitivity and specificity of the assays in a clinical setting. Additionally, compare results from patient samples with those of healthy controls to establish baseline variability and assess this assays' diagnostic potential.

Unfortunately, PBMC samples from patients with known Complex V mutations are currently not available to us. However, we have now included an in-gel Complex V activity image of cultured dermal fibroblast extracts from four control subjects and a patient with a mitochondrial translation defect resulting in a Complex V deficiency as a supplementary figure (S6 Fig).

In addition, we have increased the number of control samples used in all assays from 2 to 6 to establish baseline variability.

2. The authors need to assess Complex V functionality under stress conditions, PBMCs should be treated with metabolic stressors such as glucose deprivation or hypoxia. Furthermore, investigating whether changes in mitochondrial membrane potential (ΔΨm) correlate with ATPase or ATP synthase activity under these conditions will provide deeper insights into the enzyme's adaptability and response to metabolic challenges.

We have already assessed Complex V functionality under stressed conditions by culturing the PMBCs for 3 d in the presence of increasing concentrations of oligomycin A to validate our assays. We do not think that glucose deprivation or hypoxia will add to the validation of our assays.

Furthermore, we consider that investigating whether changes in mitochondrial membrane potential (ΔΨm) correlate with ATPase or ATP synthase activity under glucose deprivation or hypoxic conditions to provide deeper insights into the enzyme's adaptability and response to metabolic challenges is outside the scope of the current manuscript. The objective of our study is to report novel, small-scale protocols to characterize Complex V activity and assembly in PBMCs to investigate if PBMCs can serve as a platform for minimally invasive investigations of patients suspected of Complex V deficiency or in biomarker research of mitochondrial function. The objective was not to provide deeper insights into the enzyme's adaptability and response to metabolic challenges.

3. Utilizing a Seahorse XF Analyzer is essential to evaluate mitochondrial bioenergetics, including ATP production, basal respiration, and spare respiratory capacity. These analyses will offer a more comprehensive understanding of Complex V functionality in PBMCs.

As mentioned above, the objective of our study is to report novel, small-scale protocols to characterize Complex V activity and assembly in PBMCs. The objective was not to offer a more comprehensive understanding of Complex V functionality in PBMCs. Besides, the Seahorse XF Analyzer does not evaluate ATP production, it evaluates mitochondrial respiration (oxygen consumption) linked to ATP production.

4. The assays should be tested in other blood-derived cells, such as monocytes, platelets, and lymphocytes, to determine whether they are specific to PBMCs or applicable across different cell types. Additionally, extending testing to non-blood-derived cells, such as fibroblasts, will confirm the versatility of the assays.

We have chosen to use PBMCs to provide a protocol for the functional characterization of Complex V using a readily available cell type in a clinical setting. However, we have now included an in-gel Complex V activity image of cultured dermal fibroblast extracts to confirm versatility of the assay (S6 Fig).

5. Measuring reactive oxygen species production and correlating these findings with Complex V dysfunction and ΔΨm changes will provide valuable insights into the broader implications of mitochondrial dysfunction and oxidative stress.

As mentioned earlier, the objective of our study is to report novel, small-scale protocols to characterize Complex V activity and assembly in PBMCs. The objective was not to correlate reactive oxygen species production with Complex V dysfunction and ΔΨm changes.

Reviewer #2: The authors developed small-scale assays for evaluating Complex V activity and assembly using as few as 2 million PBMCs, making the process less invasive and feasible with limited blood samples. This is a significant advancement for diagnostic and research applications, especially for diseases involving mitochondrial dysfunction.

This manuscript is easy to read, and the logic is clear and understandable. The experimental approaches are appropriate and provide data to support these findings. However, this work still has a small sample size. Therefore, the reviewer is supportive of its publication until the authors address several aspects as follow:

We thank Reviewer 2 for the time and consideration to review our manuscript, and the sensible suggestions to improve it.

Major issue

Major issue 1: Limited Sample Size. All the experiments seem to have only 2 replicates (n=2) (there is even a group of data without an error bar in Figure 3). Figure 1, Figure 4, and Figures S1-3 appear to have only n=1. More replicates are needed to strengthen the experimental conclusions. Please ensure that n≥3.

We fully agree with reviewer 2 that the limited sample size was a major issue in our original manuscript. Therefore, we have now increased the number of replicates from 2 to 6 (n=6) to strengthen the conclusions of the manuscript. Figs 2, 3 and 5 have been revised. In the new figures, data points of corresponding PMBC cultures are connected and, following statistical analyses, P<0.05 values have been indicated. The additional experiments confirmed our earlier work with n=2 and our conclusions have not changed.

Furthermore, current work did not take into account other factors, such as age and sex, as well as patients with variants in Complex V genes. Including more results with this information would make the authors' small-scale assays more robust and reliable for future potential diagnostic applications.

We agree with Reviewer 2 that information regarding age and sex of the donors should have been provided in our original manuscript. We have now added a supplementary table with the age and sex of the blood donors (S Table). Sadly, PBMC samples from patients with variants in Complex V genes are currently not available to us. However, we have now included a Complex V in-gel activity image of cultured dermal fibroblast extracts from four control subjects and a patient with a mitochondrial translation defect resulting in a Complex V deficiency as a supplementary figure (S6 Fig).

Major issue 2: Concentration Determination. In the preliminary titration experiment (Figure S1), were these concentrations chosen randomly? Why were more concentrations around 62.5 nM not tested, such as 50-70 nM? The reviewer doubts that another concentration, rather than 62.5 nM, may give the maximum inhibition. A similar question arises regarding Figure S2; why were more concentrations around 0.01% not tested, such as 0.005-0.015%? The reviewer doubts that another digitonin concentration, rather than 0.01%, might result in the maximum ATPase activity.

We agree with Reviewer 2 that the determination of the optimal concentrations of oligomycin A and digitonin requires a more detailed analysis. Therefore, we have repeated the experiments to narrow down the optimal concentrations. The results are shown in the revised supplementary figures (S1 Fig and S2 Fig) and confirm the concentrations of oligomycin A and digitonin chosen in our original manuscript.

Major issue 3: The Optimal Number of PBMCs. Lines 117-120 state: 'Addition of 62.5 nM oligomycin A resulted in a decrease of ATPase activity of 12%, 39%, 39%, and 41% when 8 × 10^5, 4 × 10^5, 2 × 10^5, or 1 × 10^5 PBMCs were used, respectively (Figure S3). We chose 4 × 10^5 PBMCs as the optimal number, considering its relatively high total ATPase activity and percentage inhibition by oligomycin A.' The group of 2 × 10^5 PBMCs also gives results very similar to those of the group of 4 × 10^5 PBMCs. Considering that this paper aims to establish a protocol using fewer PBMCs, why not choose 2 × 10^5 PBMCs as the optimal number?

We agree that the relatively high total ATPase activity and percentage inhibition by oligomycin A for PMBC numbers between 1–4 × 105 are quite similar. We suggest using 4 × 105 PBMCs because the isolation of 4 × 105 PBMCs requires only a little bit more blood than the isolation of 2 × 105 PBMCs and when PBMCs from patients with a Complex V deficiency will be evaluated in future experiments, the activity will be lower than in control samples and might become undetectable when <4 × 105 PBMCs are used.

Major issue 4: Discrepancy Between Figures. Figure 5 seems to have an opposite conclusion to Figure S3. In Figure S3, the oligomycin A decrease effect on ATPase activity was attenuated when the PBMC cell number increased. However, Figure 5 shows that a larger PBMC cell number strengthens the oligomycin A decrease effect on ATPase activity. How do the authors explain the discrepancy between the conclusions of these two figures?

We do not think that there is a discrepancy between S3 Fig and Fig 5. S3 Fig shows that the total ATPase activity as well as the oligomycin A sensitive ATPase activity increase progressively with increasing numbers of PBMCs. In the experiment shown in S3 Fig, oligomycin A (acting as Complex V inhibitor) was present in sister wells of the 96-well plate during the spectrophotometric activity assay (see S7 Fig) to determine the oligomycin A sensitive ATPase activity (=CX-V activity). In Fig 5, however, PMBCs were cultured for 3 d in the presence of increasing concentrations of oligomycin A (acting as stressor), followed by in-gel Complex V activity staining in the absence of oligomycin. Throughout the manuscript, the text has been clarified by indicating where oligomycin A is used as a stressor during 3-d culturing and where as an inhibitor during the measurements.

Minor issue

Minor issue 1: Lines 59-60, “Human CX-V is comprised of 18 protein subunits. Two subunits are encoded on the mitochondrial DNA, while the remaining subunits are encoded on the nuclear DNA.” Please provide the specific names of the two subunits. Additionally, one of John E. Walker's papers should be cited when discussing the 18 subunits: https://doi.org/10.1073/pnas.1722086115 or https://doi.org/10.1039/9781788010405-00338.

We have now provided the names of the two mtDNA-encoded subunits (MTATP6 and -8) and included a reference [2] to Walker (2017).

Minor issue 2: Be more specific instead of using general descriptors.

Provide one or two specific names in lines 60-61: “Moreover, several nuclear-encoded assembly factors are required to build a functional CX-V [7].”

Provide one or two specific defect/disorder symptoms or names in lines 62-63: “They are associated with severe metabolic defects and neurodegenerative disorders [8].”

Provide one or two specific gene names in lines 64-66: “More recently, mutations in several nuclear genes involved in CX-V biosynthesis have been identified, and since the advent of next-generation sequencing, their number is rapidly increasing (reviewed in: [7]).”

Provide one or two specific disease names in line 68: “Because CX-V deficiency is increasingly recognized as a cause of disease.”

We have now provided names of nuclear-encoded assembly factors (ATPAF1, ATPAF2, FMC1, TMEM70 and TMEM242) that are required to build a functional CX-V.

We have now provided specific defect/disorder symptoms caused by Complex V deficiency.

We have now provided specific names (ATP5F1E, ATPAF2 and TMEM70) and literature references of three mutated genes.

We have not provided one or two specific disease names in the sentence starting with: “Because CX-V deficiency is increasingly recognized as a cause of disease” as the beginning of this sentence summarises the conclusion of the preceding paragraph and specific disease names are (now) given in the preceding paragraph.

When specifying these sentences above, please add reference(s) if needed.

References [13–15] were added were appropriate.

Minor issue 3: Line 122 states 'with 62.6 nM oligomycin A.' Is this a typo? Why does it use a different concentration of 62.6 nM, rather than the 62.5 nM mentioned in other contexts?

Typo has been corrected.

Minor issue 4: The figure legends for Figures 2 and 3 are ambiguous.

In Figure 2, “the percentage of oligomycin A-sensitive CX-V activity relative to total ATPase activity in samples not treated with 62.5 nM oligomycin A during the measurements (n=2).” How does the total ATPase activity of each group come about? Please specify whether it comes from the PBMCs cultured for 3 days without oligomycin A treatment, or from the fresh PBMCs (cultured for 0 days) without oligomycin A treatment. If it comes from the former, how is the first group's relative value calculated (since it is already the 'control group' without oligomycin A treatment)? In addition to figure S5, please describe the experimental protocol for Figure 2 in more detail in the Methods section. Additionally, it seems unnecessary to mention the 62.5 nM concentration in the figure legend, which is actually 0.

In Figure 3, “the percentage of gated cells treated with the indicated toxins during the measurements compared to the gated cells not treated during the measurements (n=2).” For each test group, does “the gated cells not treated during the measurements” refer to PBMCs cultured for 3 days or fresh PBMCs? Please also describe the experimental protocol for Figure 3 in more detail in the Methods section.

Oligomycin A was used as inhibitor during the spectrophotometric measurements to determine the oligomycin A sensitive ATPase activity, which corresponds to the Complex V activity. During the spectrophotometric activity measurements, each sample was measured twice, once without and once with 62.5 nM oligomycin A (see S7 Fig), to determine the percentage oligomycin A sensitive activity relative to the total ATPase activity measured in wells without oligomycin A. In addition, oligomycin A was used as a stressor during 3-d culturing of the PBMCs to induce Complex V deficiency and validate our assays. After harvesting of these cells, spectrophotometric assays were carried out of each sample without and with oligomycin A to determine the percentage oligomycin-sensitive ATPase activity in non-stressed cells (0 nM oligomycin A for 3 d) and stressed cells (1–100 nM oligomycin A for 3 d). This is now more clearly described in the text and figure legends.

Likewise, in the experiments shown in Fig 3, oligomycin A was used as a stressor during 3-d culturing of the PBMCs to induce Complex V deficiency and validate our assays. After 3 d, flow cytometric measurements were performed without or with oligomycin A, without or with rotenone + antimycin A, or without or with FCCP to determine the acute effect of oligomycin A, rotenone + antimycin A and FCCP on ��m in non-stressed cells (cultured in 0 nM oligomycin A for 3 d) and stressed cells (cultured in 1–100 nM oligomycin A for 3 d) (see S7 Fig). This is now more clearly described in the text and figure legends.

Minor issue 5: Lines 146-148 state, “Addition of oligomycin A during the TMRM staining of PBMCs resulted in a ~33% increase of gated cells (i.e., cells with a high TMRM staining intensity) compared to cells stained in absence of oligomycin A (Fig 3).” Does this sentence refer to the group colored in white? Actually, the white group is treated without oligomycin A in Figure 3.

This refers to cells that were not stressed during the 3-d culturing (without oligomycin A). This is no

---

## [Decision Letter · Decision Letter 1]

2 Apr 2025

Small-scale protocols to characterize mitochondrial Complex V activity and assembly in peripheral blood mononuclear cells

PONE-D-25-01588R1

Dear Dr. Taanman,

We’re pleased to inform you that your manuscript has been judged scientifically suitable for publication and will be formally accepted for publication once it meets all outstanding technical requirements.

Kind regards,

Xiaosheng Tan

Academic Editor

PLOS ONE

Additional Editor Comments (optional):

Reviewers' comments:

Reviewer's Responses to Questions

**Comments to the Author**

1. If the authors have adequately addressed your comments raised in a previous round of review and you feel that this manuscript is now acceptable for publication, you may indicate that here to bypass the “Comments to the Author” section, enter your conflict of interest statement in the “Confidential to Editor” section, and submit your "Accept" recommendation.

Reviewer #1: All comments have been addressed

Reviewer #2: All comments have been addressed

Reviewer #3: All comments have been addressed

2. Is the manuscript technically sound, and do the data support the conclusions?

Reviewer #1: Yes

Reviewer #2: Yes

Reviewer #3: Yes

3. Has the statistical analysis been performed appropriately and rigorously? 

Reviewer #1: Yes

Reviewer #2: Yes

Reviewer #3: Yes

4. Have the authors made all data underlying the findings in their manuscript fully available?

Reviewer #1: Yes

Reviewer #2: Yes

Reviewer #3: Yes

5. Is the manuscript presented in an intelligible fashion and written in standard English?

Reviewer #1: Yes

Reviewer #2: Yes

Reviewer #3: Yes

6. Review Comments to the Author

Reviewer #1: Most of my concerns have been adequately addressed, and I believe this paper is now in a strong position to be accepted.

Reviewer #2: The authors addressed issues in current version compared to the original version. The reviewer supports this manuscript for publication.

Reviewer #3: The author address all my concern. They improved their manuscript and data. It is acceptable for publication.

7. PLOS authors have the option to publish the peer review history of their article (what does this mean? ). If published, this will include your full peer review and any attached files.

**Do you want your identity to be public for this peer review?** For information about this choice, including consent withdrawal, please see our Privacy Policy .

Reviewer #1: **Yes: ** Jing Ju

Reviewer #2: No

Reviewer #3: No

---

## [Editor Report · Acceptance letter]

PONE-D-25-01588R1

PLOS ONE

Dear Dr. Taanman,

I'm pleased to inform you that your manuscript has been deemed suitable for publication in PLOS ONE. Congratulations! Your manuscript is now being handed over to our production team.

Kind regards,

on behalf of

Dr. Xiaosheng Tan

Academic Editor

PLOS ONE